# DayDreamer: World Models for Physical Robot Learning

**Philipp Wu\***    **Alejandro Escontrela\***    **Danijar Hafner\***

**Ken Goldberg**    **Pieter Abbeel**

University of California, Berkeley

*Equal contribution

**Abstract:** To solve tasks in complex environments, robots need to learn from experience. Deep reinforcement learning is a common approach to robot learning but requires a large amount of trial and error to learn, limiting its deployment in the physical world. As a consequence, many advances in robot learning rely on simulators. On the other hand, learning inside of simulators fails to capture the complexity of the real world, is prone to simulator inaccuracies, and the resulting behaviors do not adapt to changes in the world. The Dreamer algorithm has recently shown great promise for learning from small amounts of interaction by planning within a learned world model, outperforming pure reinforcement learning in video games. Learning a world model to predict the outcomes of potential actions enables planning in imagination, reducing the amount of trial and error needed in the real environment. However, it is unknown whether Dreamer can facilitate faster learning on physical robots. In this paper, we apply Dreamer to 4 robots to learn online and directly in the real world, without any simulators. Dreamer trains a quadruped robot to roll off its back, stand up, and walk from scratch and without resets in only 1 hour. We then push the robot and find that Dreamer adapts within 10 minutes to withstand perturbations or quickly roll over and stand back up. On two different robotic arms, Dreamer learns to pick and place objects from camera images and sparse rewards, approaching human-level teleoperation performance. On a wheeled robot, Dreamer learns to navigate to a goal position purely from camera images, automatically resolving ambiguity about the robot orientation. Using the same hyperparameters across all experiments, we find that Dreamer is capable of online learning in the real world, which establishes a strong baseline. We release our infrastructure for future applications of world models to robot learning. Videos are available on the project website: https://danijar.com/daydreamer

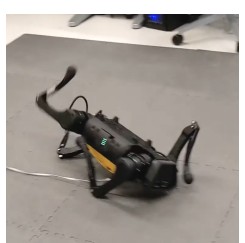   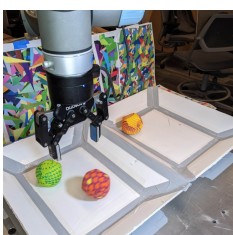   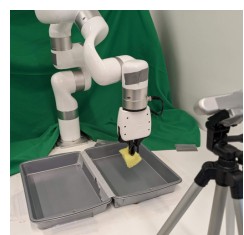   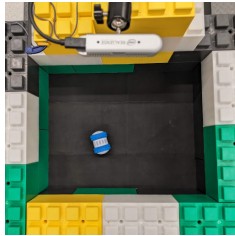

(a) A1 Quadruped Walking    (b) UR5 Visual Pick Place    (c) XArm Visual Pick Place    (d) Sphero Navigation

Figure 1: To study the applicability of Dreamer for sample-efficient robot learning, we apply the algorithm to learn robot locomotion, manipulation, and navigation tasks from scratch in the real world on 4 robots, without simulators. The tasks evaluate a diverse range of challenges, including continuous and discrete actions, dense and sparse rewards, proprioceptive and camera inputs, as well as sensor fusion of multiple input modalities. Learning successfully using the same hyperparameters across all experiments, Dreamer establishes a strong baseline for real world robot learning.

6th Conference on Robot Learning (CoRL 2022), Auckland, New Zealand.

# 1 Introduction

Teaching robots to solve complex tasks in the real world is a foundational problem of robotics research. Deep reinforcement learning (RL) offers a popular approach to robot learning that enables robots to improve their behavior over time through trial and error. However, current algorithms require too much interaction with the environment to learn successful behaviors. Recently, modern *world models* have shown great promise for data efficient learning in simulated domains and video games (Hafner et al., 2019; 2020). Learning world models from past experience enables robots to imagine the future outcomes of potential actions, reducing the amount of trial and error in the real environment needed to learn.

While learning accurate world models can be challenging, they offer compelling properties for robot learning. By predicting future outcomes, world models allow for planning and behavior learning given only small amounts of real world interaction (Gal et al., 2016; Ebert et al., 2018). Moreover, world models summarize general dynamics knowledge about the environment that, once learned, could be reused for a wide range of downstream tasks (Sekar et al., 2020). World models also learn representations that fuse multiple sensor modalities and integrate them into latent states, reducing the need for sophisticated state estimators. Finally, world models generalize well from available offline data (Yu et al., 2021), which further accelerates learning in the real world.

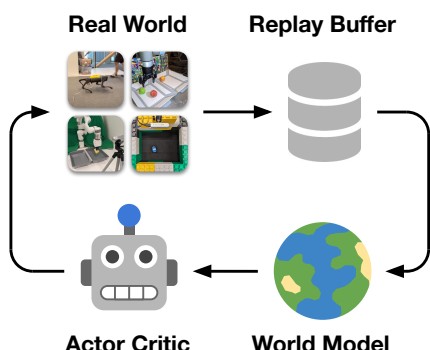

Figure 2: Dreamer follows a simple pipeline for online learning on robot hardware without simulators. The current learned policy collects experience on the robot. This experience is added to the replay buffer. The world model is trained on replayed off-policy sequences through supervised learning. An actor critic algorithm optimizes a neural network policy from imagined rollouts in the latent space of the world model. We parallelize data collection and neural network learning.

Despite the promises of world models, learning accurate world models for the real world is a open challenge. In this paper, we leverage recent advances of the Dreamer world model for training a variety of robots in the most straight-forward and fundamental problem setting: online reinforcement learning in the real world, without simulators or demonstrations. As shown in Figure 2, Dreamer learns a world model from a replay buffer of past experience, learns behaviors from rollouts imagined in the latent space of the world model, and continuously interacts with the environment to explore and improve its behaviors. Our aim is to push the limits of robot learning directly in the real world and offer a robust platform to enable future work that develops the benefits of world models for robot learning. The key contributions of this paper are summarized as follows:

- **Dreamer on Robots**   We apply Dreamer to 4 robots, demonstrating successful learning directly in the real world, without introducing new algorithms. The tasks cover a range of challenges, including different action spaces, sensory modalities, and reward structures.

- **Walking in 1 Hour**   We teach a quadruped from scratch in the real world to roll off its back, stand up, and walk in only 1 hour. Afterwards, we find that the robot adapts to being pushed within 10 minutes, learning to withstand pushes or quickly roll over and get back on its feet.

- **Visual Pick and Place**   We train robotic arms to pick and place objects from sparse rewards, which requires localizing objects from pixels and fusing images with proprioceptive inputs. The learned behavior outperforms model-free agents and approaches the performance of a human teleoperator using the same control interface as the robot.

- **Open Source**   We publicly release the software infrastructure for all our experiments, which supports different action spaces and sensory modalities, offering a flexible platform for future research of world models for robot learning in the real world.

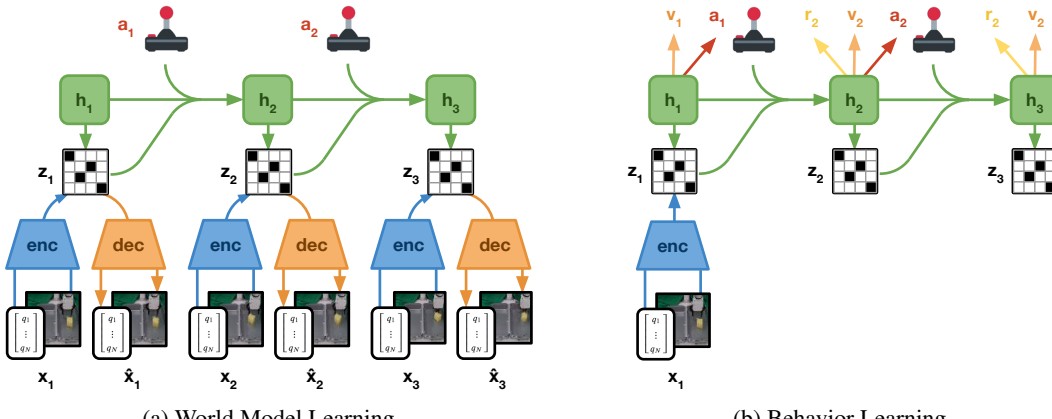

| (a) World Model Learning | (b) Behavior Learning |

**Figure 3: Neural Network Training**    We leverage the Dreamer algorithm (Hafner et al., 2019; 2020) for fast robot learning in real world. Dreamer consists of two main neural network components, the world model and the policy. **Left:** The world model follows the structure of a deep Kalman filter that is trained on subsequences drawn from the replay buffer. The **encoder** fuses all sensory modalities into discrete codes. The decoder reconstructs the inputs from the codes, providing a rich learning signal and enabling human inspection of model predictions. A **recurrent state-space model** (RSSM) is trained to predict future codes given actions, without observing intermediate inputs. **Right:** The world model enables massively parallel policy optimization from imagined rollouts in the compact latent space using a large batch size, without having to reconstruct sensory inputs. Dreamer trains a **policy network** and **value network** from the imagined rollouts and a learned **reward function**.

## 2 Approach

We leverage the Dreamer algorithm (Hafner et al., 2019; 2020) for online learning on physical robots, without the need for simulators. Figure 2 shows an overview of the approach. Dreamer learns a world model from a replay buffer of past experiences, uses an actor critic algorithm to learn behaviors from trajectories predicted by the learned model, and deploys its behavior in the environment to continuously grow the replay buffer. We decouple learning updates from data collection to meet latency requirements and to enable fast training without waiting for the environment. In our implementation, a learner thread continuously trains the world model and actor critic behavior, while an actor thread in parallel computes actions for environment interaction.

**World Model Learning**    The world model is a deep neural network that learns to predict the environment dynamics, as shown in Figure 3 (left). Because sensory inputs can be large images, we predict future representations rather than future inputs. This reduces accumulating errors and enables massively parallel training with a large batch size. Thus, the world model can be thought of as a fast simulator of the environment that the robot learns autonomously, starting from a blank slate and continuously improving its model as it explores the real world. The world model is based on the Recurrent State-Space Model (RSSM; Hafner et al., 2018), which consists of four components:

$$
\begin{array}{llll}
\text{Encoder Network:} & \mathrm{enc}_\theta(s_t \mid s_{t-1}, a_{t-1}, x_t) & \text{Decoder Network:} & \mathrm{dec}_\theta(s_t) \approx x_t \\
\text{Dynamics Network:} & \mathrm{dyn}_\theta(s_t \mid s_{t-1}, a_{t-1}) & \text{Reward Network:} & \mathrm{rew}_\theta(s_{t+1}) \approx r_t
\end{array}
\tag{1}
$$

Physical robots are often equipped with multiple sensors of different modalities, such as proprioceptive joint readings, force sensors, and high-dimensional inputs such as RGB and depth camera images. The encoder network fuses all sensory inputs $x_t$ together into the stochastic representations $z_t$. The dynamics model learns to predict the sequence of stochastic representations by using its recurrent state $h_t$. The decoder reconstructs the sensory inputs to provide a rich signal for learning representations and enables human inspection of model predictions. In our experiments, the robot has to discover task rewards by interacting with the real world, which the reward network learns to predict. Using manually specified rewards as a function of the decoded sensory inputs is also possible. We optimize all components of the world model jointly by stochastic backpropagation (Kingma and Welling, 2013; Rezende et al., 2014).

**Actor Critic Learning**   While the world model represents task-agnostic knowledge about the dynamics, the actor critic algorithm learns a behavior that is specific to the task at hand. As shown in Figure 3 (right), we learn behaviors from rollouts that are predicted in the latent space of the world model, without decoding observations. This enables massively parallel behavior learning with typical batch sizes of 16K on a single GPU. The actor critic algorithm consists of an actor network $\pi(a_t|s_t)$ and a critic network $v(s_t)$.

The role of the actor network is to learn a distribution over successful actions $a_t$ for each latent model state $s_t$ that maximizes the sum of future predicted task rewards. The critic network learns to predict the sum of future task rewards through temporal difference learning (Sutton and Barto, 2018). This allows the algorithm to take into account rewards beyond the planning horizon of $H = 16$ steps to learn long-term strategies. Given a predicted trajectory of model states, the critic is trained to regress the return of the trajectory. We compute $\lambda$-returns following Hafner et al. (2020; 2019):

$$V_t^\lambda \doteq r_t + \gamma\big((1-\lambda)v(s_{t+1}) + \lambda V_{t+1}^\lambda\big), \quad V_H^\lambda \doteq v(s_H). \tag{2}$$

While the critic network is trained to regress the $\lambda$-returns, the actor network is trained to maximize them. Different gradient estimators are available for computing the policy gradient for optimizing the actor, such as Reinforce (Williams, 1992) and the reparameterization trick (Kingma and Welling, 2013; Rezende et al., 2014) that directly backpropagates return gradients through the differentiable dynamics network (Henaff et al., 2019). Following Hafner et al. (2020), we choose reparameterization gradients for continuous control tasks and Reinforce gradients for tasks with discrete actions. In addition to maximizing returns, the actor is also incentivized to maintain high entropy to prevent collapse to a deterministic policy and maintain some amount of exploration throughout training:

$$\mathcal{L}(\pi) \doteq -\operatorname{E}\big[\textstyle\sum_{t=1}^{H} \ln \pi(a_t \mid s_t)\operatorname{sg}(V_t^\lambda - v(s_t)) + \eta\operatorname{H}\big[\pi(a_t \mid s_t)\big]\big] \tag{3}$$

We optimize the actor and critic using the Adam optimizer (Kingma and Ba, 2014). To compute the $\lambda$-returns, we use a slowly updated copy of the critic network as common in the literature (Mnih et al., 2015; Lillicrap et al., 2015). The actor and critic gradients do not affect the world model, as this would lead to incorrect and overly optimistic model predictions. The hyperparameters are listed in Appendix D.

## 3   Experiments

We evaluate Dreamer on 4 robots, each with a different task, and compare its performance to appropriate algorithmic and human baselines. The experiments are representative of common robotic tasks, such as locomotion, manipulation, and navigation. The tasks pose a diverse range of challenges, including continuous and discrete actions, dense and sparse rewards, proprioceptive and image observations, and sensor fusion. The goal of the experiments is to evaluate whether the recent successes of learned world models enables sample-efficient robot learning directly in the real world. Specifically, we aim to answer the following research questions:

- Does Dreamer enable robot learning directly in the real world, without simulators?

- Does Dreamer succeed across various robot platforms, sensory modalities, and action spaces?

- How does the data-efficiency of Dreamer compare to previous reinforcement learning algorithms?

**Implementation**   We build on the official implementation of DreamerV2 (Hafner et al., 2020). We develop an asynchronous actor and learner setup, which is essential in environments with high control rates, such as the quadruped, and also accelerates learning for slower environments, such as the robot arms. The actor thread computes online actions for the robot and sends trajectories of 128 time steps to the replay buffer. The learner thread samples data from the replay buffer, updates the world model, and optimizes the policy using imagination rollouts. Policy weights are synced from the learner to the actor every 20 seconds. We use an RSSM with 256 units to speed up the training computation. We use identical hyperparameters across all experiments, enabling off-the-shelf training on different robot embodiments.

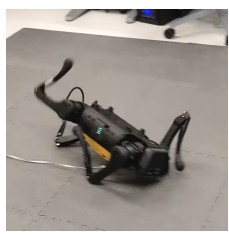 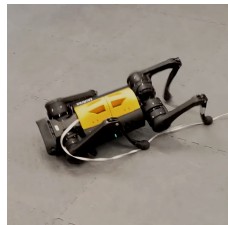 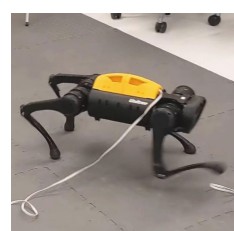 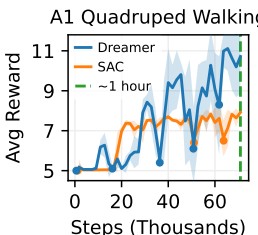

Figure 4: **A1 Quadruped Walking**   Starting from lying on its back with the feet in the air, Dreamer learns to roll over, stand up, and walk in 1 hour of real world training time, without simulators or resets. In contrast, SAC only learns to roll over but neither to stand up nor to walk. For SAC, we also had to help the robot out of a dead-locked leg configuration during training. On the right we show training curves for both SAC and Dreamer. The maximum reward is 14. The filled circles indicate times where the robot fell on its back, requiring the learning of a robust strategy for getting back up. After 1 hour of training, we start pushing the robot and find that it adapts its behavior within 10 minutes to withstand light pushes and quickly roll back on its feet for hard pushes. The graph shows a single training run with the shaded area indicating one standard deviation within each time bin.

**Baselines**   We compare to a strong learning algorithm for each of our experimental setups. The A1 quadruped robot uses continuous actions and low-dimensional inputs, allowing us to compare to SAC (Haarnoja et al., 2018a;b), a popular algorithm for data-efficient continuous control. For the visual pick and place experiments on the XArm and UR5 robots, inputs are images and proprioceptive readings and actions are discrete, suggesting algorithms from the DQN (Mnih et al., 2015) line of work as baselines. We choose Rainbow (Hessel et al., 2018) as a powerful representative of this category, an algorithm that combines many improvements of DQN. To input the proprioceptive readings, we concatenate them as broadcasted planes to the RGB channels of the image, a common practice in the literature (Schrittwieser et al., 2019). For the UR5, we additionally compare against PPO (Schulman et al., 2017), with similar modifications for fusing image and proprioceptive readings. In addition, we compare against a human operator controlling the robot arm through the robot control interface. For the Sphero navigation task, inputs are images and actions are continuous. The state-of-the-art baseline in this category is DrQv2 (Yarats et al., 2021), which uses image augmentation to increase sample-efficiency.

## 3.1   A1 Quadruped Walking

This high-dimensional continuous control task requires training a quadruped robot to roll over from its back, stand up, and walk forward at a fixed target velocity. Prior work in quadruped locomotion requires either extensive training in simulation under domain randomization, using recovery controllers to avoid unsafe states, or defining the action space as parameterized trajectory generators that restrict the space of motions (Rusu et al., 2016; Peng et al., 2018; Rudin et al., 2021; Lee et al., 2020; Yang et al., 2019). In contrast, we train in the end-to-end reinforcement learning setting directly on the robot, without simulators or resets. We use the Unitree A1 robot that consists of 12 direct drive motors. The motors are controlled at 20 Hz via continuous

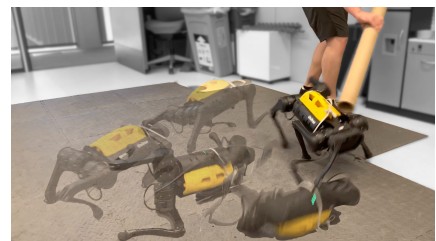

Figure 8: Within 10 minutes of perturbing the learned walking behavior, the robot adapts to withstanding pushes or quickly rolling over and back on its feet.

actions that represent motor angles that are realized by a PD controller on the hardware. Actions are filtered with a Butterworth filter to protect the motor from high-frequency actions. The input consists of motor angles, orientations, and angular velocities. Due to space constraints, we manually intervene when the robot has reached the end of the available training area, without modifying the joint configuration or orientation that the robot is in.

The reward function is the sum of five terms. An upright reward is computed from the base frame up vector $\hat{z}^T$, terms for matching the standing pose are computed from the joint angles of the hips, shoulders, and knees, and a forward velocity term is computed from the projected forward velocity $^{\mathcal{B}}v^x$ and the total velocity $^{\mathcal{B}}v$. Without the reward curriculum, the agent receives spurious reward

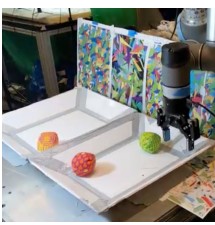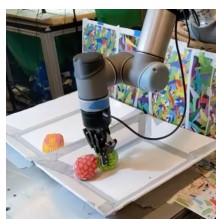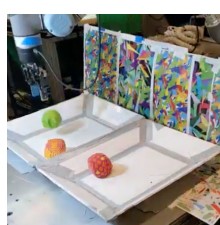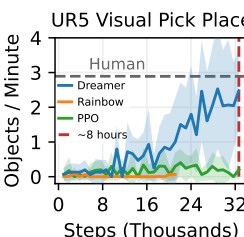

Figure 5: **UR5 Multi Object Visual Pick and Place**    This task requires learning to locate three ball objects from third-person camera images, grasp them, and move them into the other bin. The arm is free to move within and above the bins and sparse rewards are given for grasping a ball and for dropping it in the opposite bin. The environment requires the world model to learn multi-object dynamics in the real world and the sparse reward structure poses a challenge for policy optimization. Dreamer overcomes the challenges of visual localization and sparse rewards on this task, learning a successful strategy within a few hours of autonomous operation.

values due to the velocity estimator's dependence on foot-ground contact events. Each of the five terms is active while its preceding terms are satisfied to at least 0.7 and otherwise set to 0:

$$r^{\text{upr}} \doteq (\hat{z}^T[0,0,1]-1)/2 \quad r^{\text{hip}} \doteq 1 - \tfrac{1}{4}\| q^{\text{hip}} + 0.2 \|_1 \quad r^{\text{shoulder}} \doteq 1 - \tfrac{1}{4}\| q^{\text{shoulder}} + 0.2 \|_1$$
$$r^{\text{knee}} \doteq 1 - \tfrac{1}{4}\| q^{\text{knee}} - 1.0 \|_1 \quad r^{\text{velocity}} \doteq 5\big(\max(0, {}^{\mathcal{B}}v_x)/\|{}^{\mathcal{B}}v\|_2 \cdot \text{clip}({}^{\mathcal{B}}v_x/0.3, \text{-}1, 1) + 1\big) \tag{4}$$

As shown in Figure 4, after one hour of training, Dreamer learns to consistently flip the robot over from its back, stand up, and walk forward. In the first 5 minutes of training, the robot manages to roll off its back and land on its feet. 20 minutes later, it learns how to stand up on its feet. About 1 hour into training, the robot learns a pronking gait to walk forward at the desired velocity. After succeeding at this task, we tested the robustness of the algorithms by repeatedly knocking the robot off of its feet with a large pole, shown in Figure 8. Within 10 minutes of additional online learning, the robot adapts and withstand pushes or quickly rolls back on its feet. In comparison, SAC quickly learns to roll off its back but fails to stand up or walk given the small data budget.

### 3.2    UR5 Multi-Object Visual Pick and Place

Common in warehouse and logistics environments, pick and place tasks require a robot manipulator to transport items from one bin into another. Figure 5 shows a successful pick and place cycle of this task. The task is challenging because of sparse rewards, the need to infer object positions from pixels, and the challenging dynamics of multiple moving objects. The sensory inputs consist of proprioceptive readings (joint angles, gripper position, end effector Cartesian position) and a 3rd person RGB image of the scene. Successfully grasping one of the 3 objects, detected by partial gripper closure, results in a $+1$ reward, releasing the object in the same bin gives a $-1$ reward, and placing in the opposite bin gives a $+10$ reward. We control the UR5 robot from Universal Robotics at 2 Hz. Actions are discrete for moving the end effector in increments along X, Y, and Z axes and for toggling the gripper state. Movement in the Z axis is only enabled while holding an object and the gripper automatically opens once above the correct bin. We estimate human teleoperation performance by recording 3 demonstrators for 20 minutes each, controlling the UR5 with a joystick.

Dreamer reaches an average pick rate of 2.5 objects per minute within 8 hours. The robot initially struggles to learn as the reward signal is very sparse, but begins to gradually improve after 2 hours of training. The robot first learns to localize the objects and toggles the gripper when near an object. Over time, grasping becomes precise and the robot learns to push objects out of corners. Figure 5 shows the learning curves of Dreamer compared to Rainbow DQN, PPO, and the human baseline. Both Rainbow DQN and PPO only learn the short-sighted behavior of grasping and immediately dropping objects in the same bin. In contrast, Dreamer approaches human-level teleoperation performance after 8 hours. We hypothesize that Rainbow DQN and PPO fail because they require larger amounts of experience, which is not feasible for us to collect in the real world.

### 3.3    XArm Visual Pick and Place

While the UR5 robot is a high performance industrial robot, the XArm is an accessible low-cost 7 DOF manipulation, which we control at approximately 0.5 Hz. Similar to Section 3.2, the task

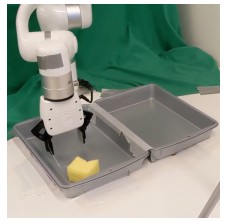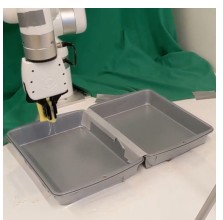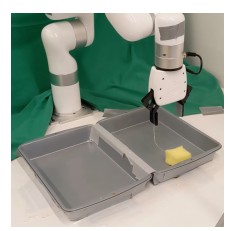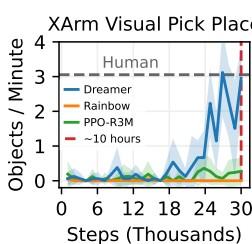

Figure 6: **XArm Visual Pick and Place**    The XArm is an affordable robot arm that operates slower than the UR5. To demonstrate successful learning on this robot, we use a third-person RealSense camera with RGB and depth modalities, as well as proprioceptive inputs for the robot arm, requiring the world model to learn sensor fusion. The pick and place task uses a soft object. While soft objects would be challenging to model accurately in a simulator, Dreamer avoids this issue by directly learning on the real robot without a simulator. While Rainbow and PPO using R3M visual embeddings converge to the local optimum of grasping and ungrasping the object in the same bin, Dreamer learns a successful pick and place policy from sparse rewards in under 10 hours.

requires localizing and grasping a soft object and moving it from one bin to another and back, shown in Figure 6. We connect the object to the gripper with a string, which makes it less likely for the object to get stuck in corners at the cost of more complex dynamics. The sparse reward, discrete action space, and observation space match the UR5 setup except for the addition of depth image observations.

Dreamer learns a policy that enables the XArm to achieve an average pick rate of 3.1 objects per minute in 10 hours of time, which is comparable to human performance on this task. Figure 6 shows that Dreamer learns to solve the task within 10 hours, whereas the Rainbow algorithm, a top model-free algorithm for discrete control from pixels, fails to learn. We additionally compare Dreamer against a PPO baseline that utilizes R3M (Nair et al., 2022) pretrained visual embeddings for the state, but notice no improvement in performance. Interestingly, we observed that Dreamer learns to sometimes use the string to pull the object out of a corner before grasping it, demonstrating multi-modal behaviors. Moreover, we observed that when lighting conditions change drastically (such as sharp shadows during sunrise), performance initially collapses but Dreamer then adapts to the changing conditions and exceeds its previous performance after a few hours of additional training, reported in Appendix A.

### 3.4 Sphero Navigation

We evaluate Dreamer on a visual navigation task that requires maneuvering a wheeled robot to a fixed goal location given only RGB images as input. We use the Sphero Ollie robot, a cylindrical robot with two controllable motors, which we control through continuous torque commands at 2 Hz. Because the robot is symmetric and the robot only has access to image observations, it has to infer the heading direction from the history of observations. The robot is provided with a dense reward equal to the negative L2 distance, which is computed using a oracle vision pipeline that detects the Sphero's position (this information is not provided to the agent). As the goal is fixed, after 100 environment steps, we end the episode and randomize the robot's position through a sequence of high power random motor actions.

In 2 hours, Dreamer learns to quickly and consistently navigate to the goal and stay near the goal for the remainder of the episode. As shown in Figure 7, Dreamer achieves an average distance to the goal of 0.15, measured in units of the area size and averaged across time steps. We find that DrQv2, a model-free algorithm specifically designed to continuous control from pixels, achieves similar performance. This result matches the simulated experiments of Yarats et al. (2021) that showed the two algorithms to perform similarly for continuous control tasks from images.

## 4   Related Work

Existing work on robot learning commonly leverages large amounts of simulated experience before deploying to the real world (Rusu et al., 2016; Peng et al., 2018; OpenAI et al., 2018; Lee et al., 2020; Irpan et al., 2020; Kumar et al., 2021; Siekmann et al., 2021; Escontrela et al., 2022), leverage fleets of robots to collect experience datasets (Kalashnikov et al., 2018; Dasari et al., 2019; Kalashnikov

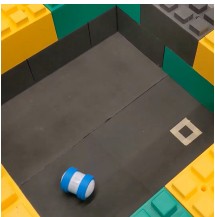 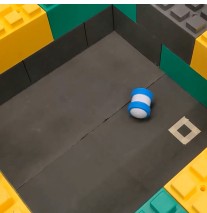 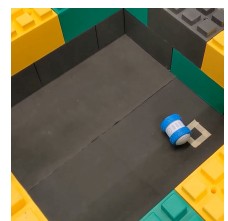 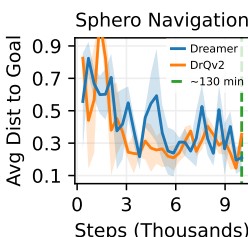

Figure 7: **Sphero Navigation**  This task requires the Sphero robot to navigate to a goal location given a top-down RGB image as the only input. The task requires the robot to localize itself from raw pixels, to infer its orientation from the sequence of past images because it is ambiguous from a single image, and to control the robot from under-actuated motors that require building up momentum over time. Dreamer learns a successful policy on this task in under 2 hours.

et al., 2021; Ebert et al., 2021), or rely on external information such as human expert demonstrations or task priors to achieve sample-efficient learning (Xie et al., 2019; Schoettler et al., 2019; James et al., 2021; Shah and Levine, 2022; Bohez et al., 2022; Sivakumar et al., 2022). However, designing simulated tasks and collecting expert demonstrations is time-consuming. Moreover, many of these approaches require specialized algorithms for leveraging offline experience, demonstrations, or simulator inaccuracies. In contrast, our experiments show that learning end-to-end from rewards in the physical world is feasible for a diverse range of tasks through world models.

Relatively few works have demonstrated end-to-end learning from scratch in the physical world. Visual Foresight (Finn et al., 2016; Finn and Levine, 2017; Ebert et al., 2018) learns a video prediction model to solve real world tasks by online planning, but is limited to short-horizon tasks and requires generating images during planning, making it computationally expensive. Yang et al. (2019; 2022) learn quadruped locomotion through a model-based approach by predicting foot placement and leveraging a domain-specific controller to achieve them. Ha et al. (2020) learn a quadruped walking policy by relying on a scripted reset policy, so the robot does not have to learn to stand up. SOLAR (Zhang et al., 2019) learns a latent dynamics model from images and demonstrates reaching and pushing with a robot arm. Nagabandi et al. (2019) learns manipulation policies by planning through a learned dynamics model from state observations. In comparison, our experiments show successful learning across 4 challenging robot tasks that cover a wide range of challenges and sensory modalities, with a single learning algorithm and hyperparameter setting.

## 5   Discussion

We applied Dreamer to physical robot learning, finding that modern world models enable sample-efficient robot learning for a range of tasks, from scratch in the real world and without simulators. We also find that the approach is generally applicable in that it can solve robot locomotion, manipulation, and navigation tasks without changing hyperparameters. Dreamer taught a quadruped robot to roll off the back, stand up, and walk in 1 hour from scratch, which previously required extensive training in simulation followed by transfer to the real world or parameterized trajectory generators and given reset policies. We also demonstrate learning to pick and place objects from pixels and sparse rewards on two robot arms in 8–10 hours.

**Limitations**  While Dreamer shows promising results, learning on hardware over many hours creates wear on robots that may require human intervention or repair. Additionally, more work is required to explore the limits of Dreamer and our baselines by training for a longer time. Finally, we see tackling more challenging tasks, potentially by combining the benefits of fast real world learning with those of simulators, as an impactful future research direction.

**Acknowledgements**  We thank Stephen James and Justin Kerr for helpful suggestions and help with printing the protective shell of the quadruped robot. We thank Ademi Adeniji for help with setting up the XArm robot and Raven Huang for help with setting up the UR5 robot. This work was supported in part by an NSF Fellowship, NSF NRI #2024675, and the Vanier Canada Graduate Scholarship.

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

# A    Adaptation

Real world robot learning faces practical challenges such as changing environmental conditions and time varying dynamics. We found that Dreamer is able to adapt to the current environmental conditions with no change to the learning algorithm. This shows promise for using Dreamer in continual learning settings (Parisi et al., 2019). Adaptation of the quadruped to external perturbations is reported in Section 3.1 and Figure 8.

The XArm, situated near large windows, is able to adapt and maintain performance under the presence of changing lighting conditions. The XArm experiments were conducted after sundown to keep the lighting conditions constant throughout training. Figure A.1 shows the learning curve of the XArm. As expected, the performance of the XArm drops during sunrise. However, the XArm is able to adapt to the change in lighting conditions in about 5 hours time and recover the original performance, which is faster than it would be to train from scratch. A careful inspection of the image observations at these times, as shown in Figure A.1, reveals that the robot received observations with strong light rays covering the scene which greatly differs from the original training observations.

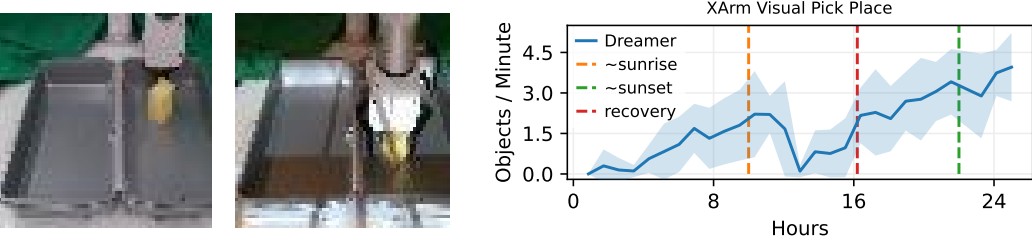

Figure A.1: The left two images are raw observations consumed by Dreamer. The leftmost image is an image observation as seen by the XArm at night, when it was trained. The next image shows an observation during sunrise. Despite the vast difference in pixel space, the XArm is able to recover, and then surpass, the original performance in approximately 5 hours. Even after 24 hours when the lighting shifts to night time conditions, the XArm is able to maintain performance.

# B    Imagination

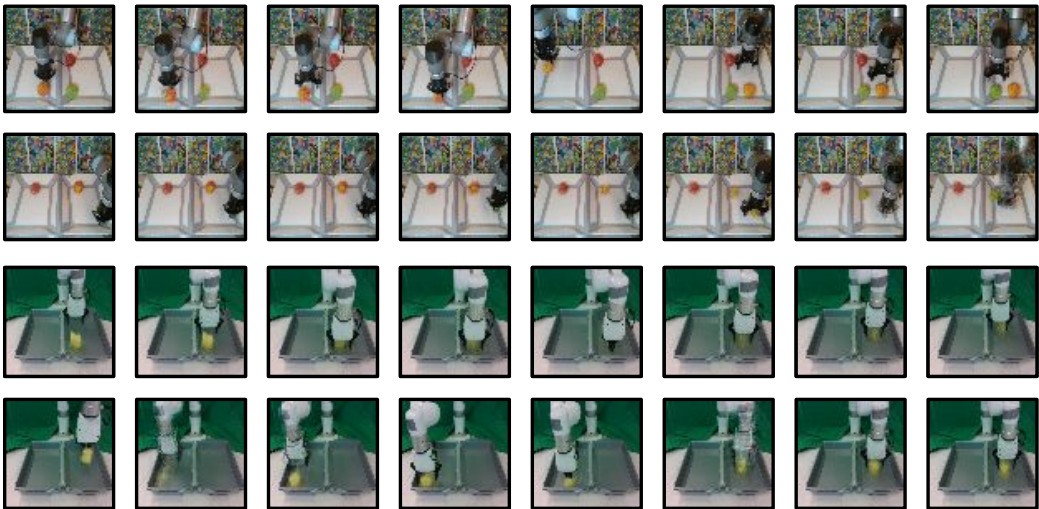

Figure B.1: To introspect the policy, we can roll out trajectories in the latent space of Dreamer, then decode the images to visualize the intent of the actor network. Each row is an imagined trajectory, showing every 2nd frame. **Top:** Latent rollouts on the UR5 environment. Multiple objects introduce more visual complexity that the network has to model. Note the second trajectory, which shows a static orange ball becoming a green ball. **Bottom:** Latent rollouts on the XArm environment.

## C  Detailed Related Work

**RL for locomotion**  A common approach is to train RL agents from large amounts of simulated data under domain and dynamics randomization (Peng et al., 2018; Lee et al., 2020; Rudin et al., 2021; Siekmann et al., 2021; Escontrela et al., 2022; Miki et al., 2022; Kumar et al., 2021; Rusu et al., 2016; Bohez et al., 2022), then freezing the learned policy and deploying it to the real world. Smith et al. (2021) explored pre-training policies in simulation and fine-tuning them with real world data. Yang et al. (2019) investigate learning a dynamics model using a multi-step loss and using model predictive control to accomplish a specified task. Yang et al. (2022) train locomotion policies in the real world but require a recovery controller trained in simulation to avoid unsafe states. In contrast, we use no simulators or reset policies and directly train on the physical robot. While prior work in locomotion has successfully learned walking behaviors in the real world, these works generally required several domain-specific assumptions or pretraining with simulators. Ha et al. (2020) achieved successful walking on the Minitaur robot in 90 minutes. However, the authors manually programmed a reset policy that was used when the robot fell on its back, while in our work the robot must learn to flip over and stand up. Additionally, the Minitaur robot is simpler than the A1 as it has 8 actuators compared to 12 on the A1. In recent work, Smith et al. (2022) utilize a high update-to-data ratio (UTD) RL algorithm to learn walking from 20 minutes of robot training data. However, their work assumes the availability of a reset policy and therefore comprises of a different learning problem compared to the problem we tackle of learning to flip over and walk from scratch. Additionally, we show our approach generalizes to environments with image observations and sparse rewards.

**RL for manipulation**  Learning promises to enable robot manipulators to solve contact rich tasks in open real world environments. One class of methods attempts to scale up experience collection through a fleet of robots (Kalashnikov et al., 2018; 2021; Ebert et al., 2021; Dasari et al., 2019; Levine et al., 2018). In contrast, we only leverage one robot, but parallelize an agent's experience by using the learned world model. Another common approach is to leverage expert demonstrations or other task priors (Pinto and Gupta, 2015; Ha and Song, 2021; Xie et al., 2019; Schoettler et al., 2019; Sivakumar et al., 2022). James and Davison (2021); James et al. (2021) leverages a few demonstrations to increase the sample-efficiency of Q learning by focusing the learner on important aspects of the scene. Other approaches, as in locomotion, first utilize a simulator, then transfer to the real world (Tzeng et al., 2015; Akkaya et al., 2019; OpenAI et al., 2018; Irpan et al., 2020). Our work focuses on single-robot environments where the agent must learn through a small amount of interaction with the world. Meanwhile, the Google Arm Farm line of work by Levine et al. leverages over 580k grasp attempts gathered by 7 robots and collected over 4 months. We believe that a method such as Dreamer could benefit greatly from this scale of training data, however it is unlikely that works such as MT-OPT/QT-OPT Kalashnikov et al. (2018; 2021) would work well in the low data regime that Dreamer excels in.

**Model-based RL**  Due to its higher sample-efficiency over model-free methods, model-based RL is a promising approach to learning on real world robots (Deisenroth et al., 2013). A model based method first learns a dynamics model, which can then be used to plan actions (Nagabandi et al., 2019; Hafner et al., 2018; Chua et al., 2018; Nagabandi et al., 2017; Becker-Ehmck et al., 2020), or be used as a simulator to learn a policy network as in Dreamer (Hafner et al., 2019; 2020). One approach to tackle the high visual complexity of the world is to learn an action conditioned video prediction model (Finn and Levine, 2017; Ebert et al., 2018; Finn et al., 2016). One downside of this approach is the need to directly predict high dimensional observations, which can be computationally inefficient and easily drift. Dreamer learns a dynamics model in a latent space, allowing more efficient rollouts and avoids relying on high quality visual reconstructions for the policy. Another line of work proposes to learn latent dynamics models without having to reconstruct inputs (Deng et al., 2021; Okada and Taniguchi, 2021; Bharadhwaj et al., 2022; Paster et al., 2021), which we see as a promising approach for supporting moving view points in cluttered environments.

# D  Hyperparameters

| Name | Symbol | Value |
|------|--------|-------|
| **General** | | |
| Replay capacity (FIFO) | — | $10^6$ |
| Start learning | — | $10^4$ |
| Batch size | $B$ | 32 |
| Batch length | $T$ | 32 |
| MLP size | — | $4 \times 512$ |
| Activation | — | $\mathrm{LayerNorm} + \mathrm{ELU}$ |
| **World Model** | | |
| RSSM size | — | 512 |
| Number of latents | — | 32 |
| Classes per latent | — | 32 |
| KL balancing | — | 0.8 |
| **Actor Critic** | | |
| Imagination horizon | $H$ | 15 |
| Discount | $\gamma$ | 0.95 |
| Return lambda | $\lambda$ | 0.95 |
| Target update interval | — | 100 |
| **All Optimizers** | | |
| Gradient clipping | — | 100 |
| Learning rate | — | $10^{-4}$ |
| Adam epsilon | $\epsilon$ | $10^{-6}$ |

# E  Environment and Hardware Details

For every robot setup that involved vision (UR5, XArm, Sphero), we used a RealSense D435 camera positioned to offer a fixed 3rd person view of the scene.

**A1**    We used the A1 quadrupedal robot by Unitree. The RL policy outputs actions at a frequency that is too high for the PD controller to track, which we overcome by lowpass filtering the action sequence. The joint range allows the legs to self-collide with the body, which can be damaging to the motors and increase battery consumption. We limited the joint range to decrease self-collisions. Finally, the EKF velocity estimator relies on foot-ground contact events to prevent significant drift in the estimates, so we employ a curriculum reward function that does not reward the robot for forward velocity until the robot is upright with extended legs. We also designed a shell which we 3D printed in order to better protect the cables and hardware and provide a smoother rolling over.

**XArm & UR5**    We utilized slanted bins to prevent objects from leaving the work area during the long-running pick and place experiments on the UR5, which is common practice Levine et al. (2018); Kalashnikov et al. (2018). We also added a partition behind the setup to keep the background constant. It would be interesting to study how a gripper-mounted camera would impact policy performance Hsu et al. (2022), however we report strong results without this design choice. For the XArm we use the uFactory xArm Gripper. For the UR5, we use the Robotiq 2F-85 parallel jaw gripper. The bin locations are predetermined and provided as part of the environment to prevent the robot from colliding with the bin. In addition, movement in the Z axis is only enabled while holding an object and the gripper automatically opens once above the other bin.

**Sphero**    We used a rectangular enclosure of $0.8 \times 0.8\mathrm{m}^2$ to keep the sphero robot within the camera view. We used a simple OpenCV script to estimate the L2 distance between the Sphero and the goal position to provide a dense reward for policy optimization. This positional information was not provided to the agent, which it had to learn from the raw top-down images.

