# OpenReview forum: "DayDreamer: World Models for Physical Robot Learning"
_robot-learning.org/CoRL/2022/Conference — CoRL 2022 Poster_

### Official Review · Reviewer_BYdE · 2022-07-30

**Originality:** Fair
**Technical Quality:** Good
**Clarity Of Presentation:** Very Good
**Impact:** 4

**Recommendation:**

Weak Accept: I recommend accepting the paper, but will not argue for my recommendation if the majority of other reviewers have a different opinion.

**Summary:**

This paper demonstrates that the previously introduced Dreamer algorithm can be applied to learning control tasks directly on real robots, without simulators. The authors successfully train policies for three different tasks: quadruped locomotion, pick and place with a robotic arm, and simple navigation of a 2-wheeled vehicle.
The Dreamer algorithm learns a world model and then uses the model to train the control policy. The purpose of this setup is to increase the sample efficiency by having the policy interact with the model instead of the real world.


**Issues:**

Contributions presented in the introduction are repetitive and can be summarized by the first point ‘Dreamer on Robots’

The sections ‘World Model Learning’ and ‘Actor-Critic Learning’ seem to be repeating [1]. There is no need to repeat this description. Instead, it would be more interesting to focus on the differences, if any.

More details should be provided about the “use of an asynchronous actor and learner” since this is one of the modifications necessary to achieve results in real life.

From my understanding, using a world model should mostly improve sample efficiency. However, the paper does not compare the efficiency of the proposed approach with more common RL algorithms such as SAC, PPO, etc.. (such an experiment could be done in simulation). It would be interesting to see how many samples these algorithms need to reach a similar performance.

The introduction claims that the world model removes the need for state estimation. However, in the quadruped example, the projected linear velocity is used to compute rewards. Isn’t it coming from the state estimator of the robot? Similarly, the pick and place task uses Forward and inverse kinematics which requires a model of the robot (of course a simple model). Finally, how is the reward of tasks 3.2, 3.3, and 3.4 computed? Don’t they require a detection pipeline?
This doesn’t hinder the usefulness of the proposed approach, but the authors should clarify that some simple models are still needed.

What new information can the reader get from experiment 3.3 compared to 3.2?

Why does SAC suffer from dead-locked leg configuration while your method doesn’t? Is this simply due to random chance or is there an explanation?

Minor issues:
In the description of figure 3, it is not clear what are the two neural networks.
Cite the work you are referring to in 3.1
The paper mentions multiple times four tasks on four robots. Three tasks on four robots seem more appropriate
Typos:
 reconstructions -> reconstructs (figure 3)
Which inapplicable -> which is inapplicable

[1]: D. Hafner, T. Lillicrap, M. Norouzi, and J. Ba. Mastering atari with discrete world models. arXiv
285 preprint arXiv:2010.02193, 2020.


**Quality Of The Limitations Section:**

Additional details required

**Reviewer Expertise:**

4: The reviewer is confident but not absolutely certain that the evaluation is correct

**Robotics Focus:**

Sufficient demonstration on hardware

**Strengths And Weaknesses:**

The main strength of the paper are the impressive results. To date, it is probably the most convincing example of a control policy trained in real life. The fact all tasks can be solved with the same hyperparameters and minimal tuning is also remarkable. Furthermore, the fact that vision based tasks can be learned live on a real robot greatly reduces the burden training such behaviours. It removes the need of complex sensor/noise modelling needed for sim2real transfer (this fact could be emphasised more in the paper). Finally, the paper is well written, structured clearly, and the supplementary video is a great addition.

On the other hand, the scientific novelty of this work is limited. While it is common (and important) to apply existing algorithms to real world tasks, such works should emphasise the main takeaways and potential modifications which were necessary to achieve the new results.


**Summary Of Recommendation:**

While there is no algorithmic novelty in this paper, the results set a new baseline for live learning on real robots. I would recommend accepting the paper if the authors extend the results section with more insights and a proper study of sample efficiency.

---

> ### Author Response · Authors · 2022-08-19
> **Rebuttal**
>
> Thank you for your review of our paper and stating that the “main strength of the paper are the impressive results. To date, it is probably the most convincing example of a control policy trained in real life”.
>
> **Repetitions**: Thank you. We will describe the Dreamer algorithm more concisely. Please refer to our official post titled *Environment Details and Hardware Specification* where we discuss decisions that facilitated continual learning in the real world.
>
> We agree with your comment "the fact that all tasks can be solved with the same hyperparameters and minimal tuning is also remarkable". Aside from the asynchronous data collection and model optimization modification, there were no changes to the original Dreamer algorithm, besides reducing the  RSSM hidden size for faster computation.
>
> **Asynchronous actor and learner**: Interleaving model optimization and policy evaluation would cause the robot to freeze during updates, so we implemented asynchronous data collection and model learning. The data collection thread computes online actions for the robot and sends trajectories of 128 time steps to the learner's replay buffer using ZMQ [1]. The learner thread reads data from the replay buffer, updates the world model, and optimizes the actor using imagination rollouts. Policy weights from the learner and data collection thread are synced every 20 seconds. Implementation details can be found in our [code](https://drive.google.com/drive/folders/1BywlUPnuUC1SAkbZ1GpgCfoiRvfo0Yv3). We will add these implementation details to the final draft.
>
> **Sample efficiency**: Thank you for raising this concern. We have created a new set of learning curve plots to make this more clear (https://imgur.com/a/TL3QgtT). We will include the plots in the final version of the paper.
>
> Work by Hafner et al. [2, 3] and Yarats et al. [4] shows extensive sample efficiency comparisons against other baselines, which we refer to for comparisons in simulated environments.
>
> **Use of simple models**: Thank you for noting the importance of simple models. We rely on the simple models and build on top of the state estimates already provided from each robot. Our intention when stating "world model removes the need for state estimation" was to note that the world model integrates information across sensors and time and thus can in principle learn state estimation itself. For example, in A1 walking tasks, while we rely on the velocity estimate provided from the A1 for the reward, Dreamer is only provided with joint angles and IMU measurements. This requires the model to implicitly learn a state estimator that measures the robot's velocity. For the Sphero navigation task, we utilize a detection pipeline, but only for reward computation. We will make these adjustments in the final draft.
>
> **Different pick & place tasks**:  The setup between the two robots is largely the same with changes to the bins and the control frequency. We leveraged the faster control rate of the UR5 and the improved bin design to run a more complex pick & place experiment with 3 potential objects to interact with and test whether the world model can deal with more complex dynamics. This is qualitatively observed through the visualized imagination rollouts in Appendix section B (in the supplemental). Imagined rollouts of the XArm have more consistency in the object dynamics than the UR5. Experimentally, we provide qualitative observations from our XArm experiments that show continual adaptation to changing lighting conditions, which is provided in Appendix A and in [this](https://imgur.com/a/FXIo3CX) additional experiment.
>
> **SAC performance**: Even though there is a dense reward incentivizing the robot to stand up and walk, the SAC policy learned to lie flat on the ground. The SAC policy did not produce actions that lead to walking, likely because they risk causing the robot to fall and enter a state with low expected return. The local optimum of lying flat was safe and had no risk of falling, allowing SAC to receive the upright reward. In comparison, Dreamer performs millions of rollouts in imagination, exploring many actions, with some that lead to walking behavior. This additional exploration within the learned world model allows the algorithm to learn walking in the real world from a small amount of interaction.
>
> **Minor issues**: Thank you for bringing our attention to these issues. We will make these adjustments accordingly.
>
> In the light of the above clarifications, we would like to ask if you are willing to increase your score assuming we have addressed your concerns. Please let us know if you have additional questions.
>
> [1] ZeroMQ, an open-source universal messaging library https://zeromq.org/
>
> [2] Dream to Control: Learning Behaviors by Latent Imagination, Hafner et al. 2020
>
> [3] Mastering Atari with Discrete World Models, Hafner et al. 2020
>
> [4] Mastering Visual Continuous Control: Improved Data-Augmented Reinforcement Learning, Yarats et al. 2021

---

### Official Review · Reviewer_VhL5 · 2022-08-02

**Originality:** Fair
**Technical Quality:** Good
**Clarity Of Presentation:** Good
**Impact:** 3

**Recommendation:**

Weak Reject: I recommend rejecting the paper, but will not argue for my recommendation if the majority of other reviewers have a different opinion.

**Summary:**

The paper evaluates the Dreamer algorithm by training directly on the hardware on 4 tasks.


**Issues:**

- see issues raised in the "weakness" section
- it is said that the "infrastructure will be released" but no link is provided nor is it in the supplementary material
- for the sphero navigation: are observation stacked?

**Quality Of The Limitations Section:**

Additional details required

**Reviewer Expertise:**

4: The reviewer is confident but not absolutely certain that the evaluation is correct

**Robotics Focus:**

Sufficient demonstration on hardware

**Strengths And Weaknesses:**

Pros:
- real world training on tasks that require use of multiple sensors.
- Dreamer manages to learn the pick and place task in the sparse reward setting
- Impressive recovering result for the quadruped robot

Cons:

One concern is the missing details for the experiments, as it is an application paper (no new method is proposed).
For instance, it is said that a low-pass filter was used but there is no information about the cut-off frequency (and how it was chosen).
Could this low-pass filter be avoided as in [2] (where training is also directly done on hardware)?
Was a history given as input, or how was the Markovian assumption not violated (see [1, 2])?

Details are also missing regarding how was the success measured, both for picking and Sphero navigation task. For instance, for the sphero task, it seems that the system can compute the distance to the target. If so, then learning probably doesn't make sense for that task.

One other concern is that the picking and sphero task are not visually challenging. Objects are easy to distinguish (good contrast with the background) and are not cluttered. The main challenge for the picking task comes from the sparse reward.

The difference between the two picking tasks is unclear. There are on two different robots, but both uses vision and proprioception and both are using objects that are relatively soft.

The recovery strategy learned is good, but the walking gait is far from something desirable (for instance, the knees touch the ground while walking), what happens if the agent is trained longer?

The baselines for the picking tasks seem unfair. One should at least provided a pre-trained (frozen) encoder to Rainbow as in [3] (where they also learn directly on hardware). For the sphero task, despite being model-free, the well-chosen baseline is competitive with the model-based approach.

Several times the paper claims to be close to "human-level performance", but this claim is misleading. It corresponds to a human operator that had access to the robot for 20 minutes.

Dreamer is meant to be data-efficient, but it sill takes 9h for the picking task...


## Remarks/Questions

- typo l224 "the the"
- the following references that train RL agent directly on real hardware seem to be missing (even from the appendix) and are probably relevant for that paper: [1, 2, 3]
- for future work, authors should consider TQC [4] (SAC with distributional critic) as model-free baseline for continuous control
- I would not spend too much time on presenting the Dreamer method and leave more space for the experiments/discussion/related work


[1] Learning to Walk via Deep Reinforcement Learning (RSS19)

[2] Smooth Exploration for Robotic Reinforcement Learning (CoRL21)

[3] Learning to Drive in a Day (ICRA19)

[4] Controlling Overestimation Bias with Truncated Mixture of Continuous Distributional Quantile Critics (ICML20)

**Summary Of Recommendation:**

I would currently recommend the paper for rejection, more experimental details are needed.

---

> ### Author Response · Authors · 2022-08-18
> **Rebuttal**
>
> Thank you for your comprehensive review of our paper. We are glad that you noted that this work led to impressive recovery behavior on the A1 and successful performance on a sparse reward task in the real world. We address your reservations below.
>
> **Environment details**: Other reviewers have asked for details regarding the environment design and hardware specifications. Please refer to the official post titled *Environment Details and Hardware Specification* for a description of our environment design.
>
> **Action filter details**: Thank you.  In the final paper we will explain that the RL policy predicts actions at a frequency that is too high for the PD controller to track. We remedy this issue by using a lowpass filter to produce a smooth action signal. The implementation details can be found [in our code](https://drive.google.com/file/d/1pfThkZ1Jra8Sw4ktbM2dWhdh1ofcnQSN/view?usp=sharing). Eliminating the need for a low pass filter would make for interesting future work.
>
> **Frame stacking**: Frame stacking was not used for Dreamer, which fuses subsequent observations into a latent state that is updated by a recurrent neural network. The resulting latent encodes a history of past observations and fuses new sensor observations. The DrQv2 baseline used frame stacking (of 5 frames) and data augmentation.
>
> **Success metrics**: For the Sphero task, we localize the center of the robot in pixel space using OpenCV for reward computation, and this information is not provided to the agent. For the pick and place task, grasp success is detected when the close action is executed but the gripper does not close fully and place success is detected by opening the gripper above the other bin; an approach often used in manipulation literature [1]. We will include more details about reward computation in the final draft.
>
> **Picking task differences**: The experimental setup between the two robots is largely the same with changes to the bins and the control frequency. The UR5 has additional complexity due to having 3 potential objects to interact with and shows that the world model can deal with more complex dynamics. This is qualitatively observed through the visualized imagination rollouts in Appendix section B (in the supplemental). Imagined rollouts of the XArm have more consistency in the object dynamics than the UR5.
>
> **Better walking behavior**: Thank you for noting that the walking behavior could display more natural properties. The knees hitting the ground is likely the result of an underspecified reward function. We chose to keep the reward function simple, while state of the art locomotion techniques use reward functions with over 10 terms [2, 3]. Future work could study how world models can be used to learn better locomotion behaviors. In [this](https://imgur.com/a/FXIo3CX) additional experiment for the xArm, we show how training for longer can improve performance and allow the world model to adapt to changes in the environment.
>
> **Additional baselines**: Thank you. Although Dreamer was not trained with a pre-trained encoder, it is interesting to compare against baselines that are aided with pre-trained visual representations. We include a run for the pick and place task on the xArm where input images are encoded with the R3M encoder [4]. We report results [here](https://imgur.com/a/hWTKmTw). Training with visual representations improves performance over the Rainbow baseline, but does not approach the performance of Dreamer.
>
> **DayDreamer v.s. human performance**: Thank you. We will update this terminology to be “performance of a human teleoperator using the same control interface as the robot”.
>
> **Data efficiency**: The pick and place task is challenging due to sparse rewards, the need to infer object positions from pixels, and the dynamics of multiple moving objects. The long training time is largely due to the low control frequency. This task takes Dreamer ~9 hours with ~22k environment steps. We believe that it is reasonable to call this data efficient. Prior work utilizes upwards of 500k grasps collected on multiple robots over a few months time [1, 2], albeit on more challenging manipulation tasks.
>
> **Remarks**: Thank you. We will fix the typos and add the missing references.
>
> In the light of the above clarifications, we hope you will consider increasing your score. Please let us know if you have additional questions.
>
> [1] Fully Autonomous Real-World Reinforcement Learning with Applications to Mobile Manipulation, Sun et al. 2021
>
> [2]: Learning Quadrupedal Locomotion over Challenging Terrain, Lee et al. 2020
>
> [3]: RMA: Rapid Motor Adaptation for Legged Robots, Kumar et al. 2021
>
> [4]: A Universal Visual Representation for Robotic Manipulation, Suraj et al. 2022
>
> [5] Learning hand-eye coordination for robotic grasping with deep learning and large-scale data collection, Levine et al. 2018
>
> [6] Qt-opt: Scalable deep reinforcement learning for vision-based robotic manipulation, Kalashnikov et al. 2018

---

> > ### Comment · Reviewer_VhL5 · 2022-08-26
> > **Re: Rebuttal**
> >
> > Thanks for your answers and clarification.
> > I have some follow-up questions.
> >
> > > Action filter details
> >
> > How did you tune it, how long did it take? What is the cut-off frequency?
> >
> > > Frame stacking
> >
> > thanks for the clarification. No frame-stacking or at least feeding previous action was done for SAC on the walking task?
> > If not, because of the use of low-pass filter (and probably communication delays), it would break Markov assumption...
> > (looking at the code, you also need to do infinite bootstrap for SAC on human intervention if it's not a failure, for instance
> > when leaving the tracking limits)
> >
> > > a simple OpenCV script was used to estimate the L2 distance between the Sphero and the goal position
> >
> > Why not provide that information to the agent?
> > I assume that by using the difference in position between two frames you could detect the orientation?
> > It sounds like learning is not needed (or at least learning from pixels).
> >
> > > grasp success is detected when the close action is executed but the gripper does not close fully and place success is detected by opening the gripper above the other bin;
> >
> > Looking at the task, once the grasp is successful, because the bin is not cluttered and you know the position of the other bin,
> > is learning the place task really needed?
> >
> > > We include a run for the pick and place task on the xArm where input images are encoded with the R3M encoder [4].
> >
> > I appreciate this additional experiment. At the same time, why did you choose PPO and not RAINBOW? (PPO is known to be less sample efficient, which would make the comparison unfair again)
> > Also R3M is a valid choice, but a simple denoising auto-encoder (pre-trained using random or manually exploration) should also be closer to what DREAMER uses, no? (if I understand well, it uses a reconstruction loss to encode the observation)
> >
> > > The long training time is largely due to the low control frequency.
> >
> > why such a low control frequency is needed? Did you try with higher in preliminary experiments?

---

> > > ### Author Response · Authors · 2022-08-28
> > > **Response**
> > >
> > > **Action filter details**:
> > > No tuning was done on the action filter. We use this value based on prior work that also performs learning with the A1. Better performance may be possible by additional tuning.
> > >
> > > **Frame Stacking**:
> > > Thank you for your helpful feedback. Infinite bootstrapping is performed after manual interventions for the SAC policy, as in Dreamer (line 129 in pytorch_sac/train.py). No frame stacking was used for Dreamer or the SAC agent.
> > >
> > > **Position information for the sphero task**:
> > > Position information was not provided to the agent since we wanted this task to only take image observations. It is true that this task can be solved without learning, but may require additional domain knowledge such as developing a model of the system dynamics. The goal was to show that this contrived task *can* be solved with learning.
> > >
> > > **Place task**: This is correct, given a successful pick, a successful place can be automatically scripted. This is not a design decision we decided to make. We instead chose to limit the amount of expert domain knowledge provided to the learner.
> > >
> > > **Using R3M for PPO** : We choose to use R3M as:
> > > R3M has been shown to be an effective pretrained model for extracting visual representation for control and shown success on real world manipulation tasks [1].
> > > Given the limited time to integrate and run the experiments, it was straightforward to integrate.
> > > We leave the potential for using a denoising auto-encoder or other potential representation learning methods for future work, as this is not the focus of the work.
> > >
> > > The reason for using PPO was that from our initial experiments, we found PPO to learn slightly better than RAINBOW. See our updated plots under UR5 Visual Pick and Place where we have PPO and RAINBOW as baselines (https://imgur.com/a/TL3QgtT). The reason for using PPO was because it showed better performance when compared to RAINBOW.
> > >
> > > **Control frequency for manipulation experiments**: The design choice we made for these experiments was to keep everything synchronous, so observations at any given time are reflective of the current state. As such we were limited by the execution time for the robot, specifically the gripper open and close actions which takes ~2 seconds on the XArm. We reported the approximate average control frequency. Due to this limitation we were unable to try high frequencies.
> > >
> > > We would also like to emphasize that one of our main contributions that we are trying to share is showing that using world models (specifically Dreamer) can be used to learn directly in the real world across a wide range of tasks and modalities including locomotion and object pick and place. The goal of the environments was to show feasibility on various control tasks (ie, continuous control on states, discrete control with images inputs, continuous control on images), even if some like the Sphero Navigation task are a bit contrived in nature. Solving any one specific task may be done more optimally, but Dreamer is able to accomplish all tasks with (arguably) less task specific environment and control design, and opens the possibility of use for much more difficult robotics tasks in the future
> > >
> > > Please let us know if anything else can be clarified.
> > >
> > > [1]: A Universal Visual Representation for Robotic Manipulation, Suraj et al. 2022

---

### Official Review · Reviewer_HuH8 · 2022-08-03

**Originality:** Good
**Technical Quality:** Good
**Clarity Of Presentation:** Very Good
**Impact:** 4

**Recommendation:**

Weak Accept: I recommend accepting the paper, but will not argue for my recommendation if the majority of other reviewers have a different opinion.

**Summary:**

In this paper, the authors applied the Dreamer algorithm directly on real robots to learn locomotion, manipulation and navigation tasks. Except for the parallel execution of data collection and controller learning, the paper made little change to Dreamer and applied it as-is. Nevertheless, the tasks were trained within a reasonable time budget. The authors therefore demonstrated the feasibility of applying a model-based RL algorithm for online learning on physical robots and established baselines for future methods.

**Issues:**

* Was the parallel data collection and learning also applied to the baselines? The reviewer assumes this is not the case due to the nature of the baselines. If not, Dreamer may have consumed more data. What is the amount of data used for training in Dreamer and the baselines?
* Can the authors add the specification of the hardware used for learning?
* DrQv2 seems promising too. If data augmentation were applied to the baselines in other experiments (e.g. the pick and place tasks), what would the performance gaps look like? How narrow would they be?

**Quality Of The Limitations Section:**

Limitations are addressed clearly

**Reviewer Expertise:**

3: The reviewer is fairly confident that the evaluation is correct

**Robotics Focus:**

Sufficient demonstration on hardware

**Strengths And Weaknesses:**

Strengths
* The idea of applying Dreamer directly on robots to learn real-world world model is novel.
* The results cover a variety of robotic tasks. They encourage future research that explore/improve model-based RL methods directly on physical robots and serve as baselines.

Weaknesses
* Lack of details to support data efficiency. While wall clock time comparison is an important criteria, the number of trajectories/data used for learning should also be put to check.
* Lack of hardware setup info. While training a locomotion controller on a physical robot within 1 hour is impressive, the readers may also want to know the spec of the hardware that allowed this.


**Summary Of Recommendation:**

The paper is well written and easy to follow. Although the technical innovation is mainly applying Dreamer directly to physical robots, the results are encouraging, and will likely inspire more future works along the same direction. The reviewer believes the paper can be improved if some details and additional analysis (see Issues below) are added.

---

> ### Author Response · Authors · 2022-08-18
> **Rebuttal**
>
> Thank you for your detailed review of our paper and for noting that “The results cover a variety of robotic tasks. They encourage future research that explore/improve model-based RL methods directly on physical robots and serve as baselines.”
>
> We appreciate your points with regards to Weaknesses and Issues, and address them below:
>
> **Lack of details to support data efficiency**: Thank you for raising this and we apologize for the lack of clarity. The current learning curves we show actually reflect the number of data samples, but was rescaled according to our average control frequency to reflect training time. We have created a new set of learning curve plots to make this more clear (https://imgur.com/a/TL3QgtT), and will update the plots in the final version of the paper.
>
> **Lack of hardware info**: We will clarify the physical robot setup in the appendix of the final version. For the quadruped, we used the A1 from unitree [1]. We can include a citation to the user manual which provides the exact hardware specifications. In addition to the off the shelf A1 quadruped, we also designed a shell which we 3D printed in order to better protect the cables and hardware and provide a smoother surface for the robot to roll over from. For the manipulation experiments, we utilized slanted bins to prevent objects from leaving the work area while still allowing for the arm to traverse the workspace on the UR5.The bin locations are pre determined and provided as part of the environment to prevent the robot from colliding with the bin. We also added a partition behind the arms to keep the background constant. For the XArm we use the uFactory xArm Gripper [2]. For the UR5, we use the Robotiq 2F-85 parallel jaw gripper [3]. For the Sphero robot experiment, we built a rectangular enclosure of approximately 0.8x0.8 m^2 which was used to keep the Sphero in the camera's view. For every robot setup that involved vision, we used a RealSense D435 camera positioned to offer a fixed 3rd person view of the scene [4]. These hardware details will be included in the final version of the paper. For training our models, we utilize an Nvidia TitanX for most experiments except for the Ur5 and A1 experiments which used an Nvidia 3090 for model optimization and an Nvidia 2080Ti for policy evaluation.
>
> **Parallel data collection with regards to baselines**: Good point.  We will address this in the final version. The parallel data collection was an additional implementation that we added. To ensure the correctness of the baselines, we used the original (or previously validated) implementations as closely as possible. Please let us know if the above plots looking at the number of training samples addresses this concern.
>
> **DrQv2 and data augmentation for other baselines**: We agree that DrQv2 displays promising results on the Sphero task. As shown in DrQv2, data augmentation can significantly increase the performance of visual RL algorithms. In future work, we will evaluate how using data augmentation or universal visual representations [5] can improve sample efficiency with learned world models for real world robotics tasks. We have run additional baselines where we encoder input images with the R3M encoder, and report those results [here](https://imgur.com/a/hWTKmTw). In addition, DrQv2 assumes a continuous action space and has only been applied to visual RL tasks, which is why we did not apply this baseline to other experiments.
>
> In the light of the above clarifications, we hope you will consider increasing your score. Please let us know if you have additional questions.
>
> [1]: Unitree A1 specifications. https://m.unitree.com/products/a1
>
> [2]: xArm gripper specifications. https://www.ufactory.cc/product-page/ufactory-xarm-gripper
>
> [3]: UR5 gripper specifications. https://robotiq.com/products/2f85-140-adaptive-robot-gripper?ref=nav_product_new_button
>
> [4]: Intel RealSense D435. https://www.intelrealsense.com/depth-camera-d435/
>
> [5]: R3M: A Universal Visual Representation for Robot Manipulation, Nair et al. 2022

---

### Official Review · Reviewer_4BTT · 2022-08-05

**Originality:** Fair
**Technical Quality:** Good
**Clarity Of Presentation:** Very Good
**Impact:** 3

**Recommendation:**

Weak Reject: I recommend rejecting the paper, but will not argue for my recommendation if the majority of other reviewers have a different opinion.

**Summary:**

The paper showcases that learning of sufficiently accurate world models for the real world is possible. In particular, the Dreamer framework is applied to four real tasks that are distinct in their particular nature. Learning is performed without using a simulator or demonstrations.

**Issues:**

It would be interesting to understand the contributions of this paper better. Why exactly are the empirical results new, since similar performances have been reported before? Please also comment on the weaknesses listed before. If addressed appropriately, I'm happy to increase the score.

**Quality Of The Limitations Section:**

Additional details required

**Reviewer Expertise:**

3: The reviewer is fairly confident that the evaluation is correct

**Robotics Focus:**

Sufficient demonstration on hardware

**Strengths And Weaknesses:**

# Strengths
- impressive performance on real robots (e.g., learn to stand up and walk on real system in 1h).
- All tasks are solved using the same algorithm and hyperparameters.
- The paper is well written despite occasional typos.
# Weaknesses
- no or very little algorithmic advances. The paper uses the existing Dreamer framework and feels more like a report in large parts of the paper. It would be nice to understand what the authors needed to do to make the Dreamer framework run on real systems compared to running it in simulated environments. Are (3) and (4) modifications to the previous work that needed to be performed for the real system?
- Contributions: Although I acknowledge that the paper is the first to make Dreamer run on real robots and the open-source argument, I doubt that this paper pioneers to learn to walk in 1h and reaches previously unseen performance on the visual pick and place task. For the letter task, the work by Sergey Levine’s group presented immense results over the past years. The results in the paper are indeed impressive, but I feel the contributions are not formulated carefully. One of the biggest contributions I see is that all tasks have been solved by a single algorithm and constant hyperparameters, which is not mentioned in this list. It would be nice if the authors would clarify if the hyperparameters of behavior learning part (Figure 3 (b)) are also the same for each task.
- The paper claims to “approach human performance” in the visual pick and place task, which implies that the robot is better at picking the objects than a human. However, in Section 3 it is mentioned that the robot is teleoperated by a human, which is a much easier baseline to beat.

**Summary Of Recommendation:**

The paper reports impressive results, but seems not to include any algorithmic advances. Moreover, the paper seems to partially overstate the performance: (i) the argument to match human performance in the pick and place task leaves room for misinterpretations and (ii) the results in the Sphero Navigation task only match the performance of the baseline.

---

> ### Author Response · Authors · 2022-08-17
> **Rebuttal**
>
> We would like to thank you for your thoughtful comments, in particular that the paper presents “impressive performance on real robots (e.g., learn to stand up and walk on real system in 1 h) and that "all tasks are solved using the same algorithm and hyperparameters”.
>
> We address your reservations regarding our paper's contributions below:
>
> **What was required to make Dreamer work in real world environments?** This is an excellent meta point that we will address in the revised version of the paper.
>
> The Dreamer algorithm was not modified between tasks, and the same hyperparameters were used across all experiments. Interleaving model optimization and policy evaluation would cause the robot to freeze during updates, so we implemented asynchronous data collection and model optimization. We also used 256 units in the RSSM hidden layer to speed up the training computation and lowered the amount of prefill data used to bootstrap model learning. We will place a strong emphasis on this fact and include these environment details in the final version of our paper since PDFs cannot be modified during the rebuttal period.
>
> Other reviewers have asked for additional details regarding the environment design and hardware specifications. Please refer to the official post titled *Environment Details and Hardware Specification* for a thorough description of our environment design. Many of these environment design decisions helped facilitate learning, using design choices consistent with other work in manipulation and locomotion.
>
> **Contributions** Thank you for noting the lack of clarity here. While prior work in locomotion has successfully learned walking behaviors in the real world, we find that these works either require more time to train, use simple robots, or pre-train with simulators. Work by Ha et al. [4] achieved successful walking on the Minitaur robot in 90 minutes. However, the authors manually programmed a reset policy that was used when the robot fell on its back, while in our work the robot must learn to flip over. Additionally, the Minitaur robot is simpler than the A1 as it has 8 actuators compared to 12 on the A1. Work by Smith et al. [5] successfully trained locomotion policies in the real world, but performs pre-training in simulation. In our work we examine learning behaviors from scratch in the real world.
>
> We agree that the work by Levine and Hausman has yielded impressive results. However, we believe that comparing DayDreamer to their line of work results in an unfair comparison. Our work focuses on single-robot environments where the agent must learn through a small amount of interaction with the world. Meanwhile, the Google Arm Farm line of work by Levine et al. leverages over 580k grasp attempts gathered by 7 robots and collected over 4 months. We believe that a method such as Dreamer could benefit greatly from this scale of training data, however it is unlikely that works such as MT-OPT/QT-OPT would work well in the low data regime that Dreamer excels in.
>
> We will update the final version of the paper to better position our results in the context of other work such as [1, 2, 4, 5, 6].
>
> **DayDreamer v.s. Human Performance**: This was poor wording on our part, thank you for pointing this out. We can update this terminology to be “performance of a human teleoperator using the same control interface as the robot”.
>
> In the light of the above clarifications, we would like to ask if you are willing to increase your score assuming we have addressed your concerns. Otherwise, please let us know if you have additional questions.
>
> [1]: Learning Hand-Eye Coordination for Robotic Grasping with Deep Learning and Large-Scale Data Collection, Levine et al. 2016
>
> [2]: QT-Opt: Scalable Deep Reinforcement Learning for Vision-Based Robotic Manipulation, Kalashnikov et al. 2018
>
> [3]: Vision-based manipulators need to also see from their hands, Hsu et al. 2022
>
> [4]: Learning to Walk in the Real World with Minimal Human Effort, Ha et al. 2020
>
> [5]: Legged Robots that Keep on Learning: Fine-Tuning Locomotion Policies in the Real World, Smith et al. 2022
>
> [6]: MT-Opt: Continuous Multi-Task Robotic Reinforcement Learning at Scale, Kalashnikov et al. 2021

---

### Official Review · Reviewer_niYi · 2022-08-10

**Originality:** Very Good
**Technical Quality:** Very Good
**Clarity Of Presentation:** Very Good
**Impact:** 4

**Recommendation:**

Weak Accept: I recommend accepting the paper, but will not argue for my recommendation if the majority of other reviewers have a different opinion.

**Summary:**

This is a system paper where DreamerV2 [Hafner 2020] was applied to real-world tasks on four different robots: A1 Quadruped walking, UR5 visual pick and place, XArm visual pick and place, and Sphero navigation. The algorithm was trained directly on the hardware and showed great performance under the same set of hyperparameters.

**Issues:**

Please see the weakness list for questions to be addressed.

**Quality Of The Limitations Section:**

Limitations are addressed clearly

**Reviewer Expertise:**

4: The reviewer is confident but not absolutely certain that the evaluation is correct

**Robotics Focus:**

Sufficient demonstration on hardware

**Strengths And Weaknesses:**

### Strengths:

1. Implementation of RL on robotics hardware

2. Great performance across 4 robots and tasks, under the same hyperparameters


### Weaknesses:

1. The author claimed software infrastructure as one contribution but it does not seem to have been included in the submission hence I can not properly assess this contribution.

2. Some details were missing that would otherwise help better understand the work:
  - (reproducibility)Line 176 mentioned the use of low-pass filter but the details were not given.
  - (reproducibility)What do the lines and the shading in the learning curves represent?
  - How does the wall clock time (1hr and 10 mins) translate to the number of samples? What was the compute resource used?
  - The manipulation experiments seem to use a green background for Xarm and a background with a mixture of colors for the UR5. Is there any particular reason for the difference in the setup?

3. Additional ablation experiments that would be interesting:
  - How was the reward function in Eq. 5 chosen? If the authors had put some efforts into reward shaping, it would be great to see an ablation study on that. If not, could the authors clarify?
  - Line 231 and Appendix A.3 showed the adaptation to lighting conditions. This is quite interesting and I’m curious to see what happens when the lighting switches back to nighttime again (after the adaptation). Could this ablation be added?

4. Minor:
  - what is sg() in Eq. 4? It should be explained.
  - Line 125 typo: reperameterization -> reparameterization


**Summary Of Recommendation:**

I applaud the authors for validating deep RL algorithms on real robots.
Running Dreamer on real-world robots is novel to my best knowledge and is potentially of high impact.
My main concerns are some clarifications and missing ablation studies that I think will strengthen the work. I will be happy to raise my score if my concerns are properly addressed.

---

> ### Author Response · Authors · 2022-08-18
> **Rebuttal**
>
> Thank you for your valuable  feedback and for stating that “Running Dreamer on real-world robots is novel to my best knowledge and is potentially of high impact”
>
> We address your questions below:
>
> **Open sourcing of the code:** Our intention was to open source before the paper deadline but we prioritized running experiments over anonymizing the code. We promise to  include a GitHub link in the final version of our paper, and you can access an anonymized version of the DayDreamer code [here](https://drive.google.com/drive/folders/1BywlUPnuUC1SAkbZ1GpgCfoiRvfo0Yv3?usp=sharing).
>
> **Clarifications:**
>
> * Low-pass filter: Thank you for noting that we didn’t sufficiently clarify this.  In the final paper we will explain that the RL policy predicts actions at a frequency that is too high for the PD controller to track. We remedy this issue by using a lowpass filter to produce a smoother action signal. Specifically, we use a butterworth filter with a high cut frequency of 1.5. The implementation details can be found [here](https://drive.google.com/file/d/1pfThkZ1Jra8Sw4ktbM2dWhdh1ofcnQSN/view?usp=sharing).
> * Shading in plot: We will add a note about this in the final version: Within each time bucket, the line represents the average reward and the shaded area represents one standard deviation.
> * Wall-clock time v.s. number of samples: We will update the final version to clarify that we run our experiments on the A1 with a control frequency of 1ms and an action repeat of 50, so the policy updates the joint angle targets ~20 times every second. At convergence, the policy had produced ~84,000 actions (0.05s/action) and the robot had performed ~4M low-level steps (0.001s/low level step). The UR5 and xArm took ~1 action per second for a period of 8 hours, resulting in about 30,000 environment interactions after 8 hours. We have produced plots where the x axis is shown in steps (https://imgur.com/a/TL3QgtT). We hope these plots clarify the relation between wall clock time and step count, and we will update the final version of the PDF to include this information.
> * Background differences for UR5 and Xarm: There was no specific reason for the different backgrounds for the UR5 and Xarm setups other than availability of background materials. Putting a background behind the arms provides cleaner training data. Without the background, Dreamer might try to model the movements of students working in the background.
>
>
> **Additional ablations:**
>
> * Adaptation to lighting conditions:
> Thank you for your interest in this ablation! We were also pleasantly surprised to see the performance quickly recover shortly after sunrise. We reran the xArm experiment as requested with an updated plot showing a full 24 hour training run. This can be viewed at this link (https://imgur.com/a/FXIo3CX). As the plot shows, the robot is not greatly impacted by the shift back to night and is able to maintain performance. This improved experiment will be updated on the final paper.
>
> * A1 reward function: We can clarify in the final version that since the EKF velocity estimator relies on foot-ground contact events to prevent significant drift in the estimates, we employ a curriculum reward function that does not reward the robot for tracking the desired velocity until the robot is upright (r_upr) and its feet are contacting the ground. In our ablations, we found that without the curriculum the agent would receive spurious rewards while the agent was on its back due to incorrect velocity estimates. Finally, the hip, knee, and shoulder rewards incentivize the robot for staying close to the nominal joint configuration, and are consistent with prior work in RL for locomotion [1][2][3].
>
> **Minor clarifications:** Thank you for pointing out these typos, we will fix all of them in the final version of our paper as the PDF cannot be updated during the rebuttal period. The "sg" operator in equation 4 is the stop gradient operation and we will clarify this in the final version.
>
> In the light of the above clarifications, we would like to ask if you are willing to increase your score assuming we have addressed your concerns. Please let us know if you have additional questions.
>
> [1]: Learning Quadrupedal Locomotion over Challenging Terrain, Lee et al. 2020
>
> [2]: RMA: Rapid Motor Adaptation for Legged Robots, Kumar et al. 2021
>
> [3]: Learning robust perceptive locomotion for quadrupedal robots in the wild, Miki et al. 2022

---

> > ### Comment · Reviewer_niYi · 2022-08-25
> > **A follow up question**
> >
> > Thank you for your clarification. I appreciate the effort to add the lighting condition ablation.
> > Could you please clarify how many runs were the mean/standard deviation based on?

---

> > > ### Author Response · Authors · 2022-08-26
> > > **Response**
> > >
> > > We ran each robot experiment once, given the relatively high cost of running robotics experiments directly in the real world. For each run, the mean and standard deviation within each time bucket is measured and plotted to provide a measure of training stability and oscillations.

---

> > > > ### Comment · Reviewer_niYi · 2022-08-27
> > > > **Regarding the non-standard setup of the learning curves**
> > > >
> > > > Thanks for clarifying what the curves in the figures really mean.
> > > >
> > > > I find it to be a non-standard setup (as opposed to the standard multiple seeds/runs) that is misleading when unspecified.
> > > > Also running experiments only once due to "relatively high cost" sort of contradicts the second contribution "walking in 1 hour" IMHO. Fig.4 and Fig. 7 show only 1 and 2 hours of training respectively. I wonder why those experiments are too expensive to perform multiple runs. Is it due to what the paper mentioned that learning on hardware "may require human intervention or repair"? If so, could the authors shed light on how long does it take to finish one run accounting for all the extra work?
> > > > Thanks!

---

> > > > > ### Author Response · Authors · 2022-08-27
> > > > > **Response**
> > > > >
> > > > > Thank you for noting the paper's limitation regarding the cost of running robot experiments. We disagree that running one experiment contradicts our claim that DayDreamer achieves walking in 1 hour without simulators. For the A1 walking task, the robot can run for a duration of about 20 minutes before the battery needs to be changed and the robot left to cool down, which pauses training. These pauses add roughly 30 minutes to the training run, resulting in ~1 hour and 30 minutes of experiment time. Additionally, while learning to stand up, the robot collides violently with the ground, which leads to significant wear on the robot. Specifically, the feet of the A1 are attached to the lower leg with a weak glue that causes the feet to unbind. A higher quality legged robot would have a longer-lasting battery, better heat management characteristics, and higher quality components that resist wear. These issues, combined with the need to run baselines, limit the amount of experiments we perform on the robot. Still, we show that only 1 hour of data gathered by running Dreamer on the robot is required to learn a policy that recovers from falls and walks forward. We agree that running the Sphero task for multiple runs is feasible. We will perform more experiments for the A1, Sphero, and XArm experiments between now and the deadline for the final version of the paper.

---

> > > > > > ### Comment · Reviewer_niYi · 2022-08-28
> > > > > > **Response**
> > > > > >
> > > > > > Thanks for the clarifications and for the plan to include more experiments.
> > > > > >
> > > > > > I think that it is important to include these details about the challenges of running RL on real robots.
> > > > > > Perhaps what I mentioned of only performing 1 run contradicting the claim of walking in 1 hr was confusing. What I meant was that being able to walk in 1 hr was impressive but such a short training time left little excuse to not include more runs. The new clarifications from the author was helpful to put into perspective how much total time it was to run this experiment.
> > > > > >
> > > > > > Let me summarize my post-rebuttal impression. I appreciate the quick responses from the authors throughout the rebuttal besides the efforts to improve the paper. My questions and concerns were addressed fairly. But I was honestly quite surprised by the non-standard setup of the learning curves and that this important piece of detail was left out in the original paper. I understand that it's generally expensive to run RL on real robots but that being said, we are talking about running *RL algorithms* and 1 run just won't pass from my view. The existing performance still looks promising. Thus I tend to maintain my score as weak accept.

---

> > > > > > > ### Author Response · Authors · 2022-08-28
> > > > > > > **Final Response**
> > > > > > >
> > > > > > > Thank you for partaking in the review process and for providing helpful feedback. We understand your decision to maintain your score, and looking forward to further improving the paper.

---

### Author Response · Authors · 2022-08-18
**Environment Details and Hardware Specification**

Careful environment design is critical for continual learning in the real world. Here we include additional details listing design choices made to facilitate learning. We also list the hardware specifications for each environment. This information will be included in the final draft of the paper.

Additionally, please refer to this [link](https://drive.google.com/drive/folders/1BywlUPnuUC1SAkbZ1GpgCfoiRvfo0Yv3) for an anonymized release of our code.

**Environment design:**

- A1: We used the A1 quadrupedal robot from Unitree [1]. The RL policy predicts actions at a frequency that is too high for the PD controller to track. We remedy this issue by using a lowpass filter to produce a smoother action signal. Additionally, the A1's joint range allows the legs to self-collide with the body, which can be damaging to the robot and lead to fast battery consumption. Limiting the joint range to decrease self-collisions decreased the amount of manual intervention required. Finally, since the EKF velocity estimator relies on foot-ground contact events to prevent significant drift in the estimates, we employ a curriculum reward function that does not reward the robot for tracking the desired velocity until the robot is upright with extended legs, which prevents spurious velocity-tracking rewards. We also designed a shell which we 3D printed in order to better protect the cables and hardware and provide a smoother surface for the robot to roll over from.

- XArm/UR5: We utilized slanted bins to prevent objects from leaving the work area during the long-running pick and place experiments on the UR5; a common practice in manipulation literature [2][3]. We also added a partition behind the arms to keep the background constant. It would be interesting to study how a gripper-mounted camera would impact policy performance [4], however we report strong results without this design choice. For the XArm we use the uFactory xArm Gripper [5]. For the UR5, we use the Robotiq 2F-85 parallel jaw gripper [6]. The bin locations are pre determined and provided as part of the environment to prevent the robot from colliding with the bin. In addition, movement in the Z axis is only enabled while holding an object and the gripper automatically opens once above the correct bin.

- Sphero: A rectangular enclosure was used to keep the sphero robot within the camera's view. Additionally, a simple OpenCV script was used to estimate the L2 distance between the Sphero and the goal position, which provided a dense reward for policy optimization. This position information was not provided to the agent, as the agent was only provided with a top-down image of the scene. We built a rectangular enclosure of approximately 0.8x0.8 m^2 which was used to keep the Sphero in the camera's view.

For every robot setup (UR5, XArm, Sphero) that involved vision, we used a RealSense D435 camera positioned to offer a fixed 3rd person view of the scene [7].

**Compute resources**:

For training our models, we utilize an Nvidia TitanX for most experiments with two exceptions. For training the world model and policy in the learner thread, we used an Nvidia 3090. For the A1 an Nvidia 2080Ti for policy evaluation.

[1]: Unitree A1 specifications. https://m.unitree.com/products/a1

[2]: Learning Hand-Eye Coordination for Robotic Grasping with Deep Learning and Large-Scale Data Collection, Levine et al. 2016

[3]: QT-Opt: Scalable Deep Reinforcement Learning for Vision-Based Robotic Manipulation, Kalashnikov et al. 2018

[4]: Vision-based manipulators need to also see from their hands, Hsu et al. 2022

[5]: xArm gripper specifications. https://www.ufactory.cc/product-page/ufactory-xarm-gripper

[6]: UR5 gripper specifications. https://robotiq.com/products/2f85-140-adaptive-robot-gripper?ref=nav_product_new_button

[7]: Intel RealSense D435. https://www.intelrealsense.com/depth-camera-d435/

---

### Meta-Review · Area_Chair_jUoM · 2022-08-09

**Recommendation:** Accept (Poster)
**Confidence:** 4

**Metareview:**

This paper demonstrated the application of the Dreamer framework to four real robot tasks. It verified that RL with world models could be applied in the real world without using simulators and demonstration data, and better than model-free RL.

Strengths:

- Impressive performance on real robots

- Four tasks are solved by the same algorithm with the same hyperparameters

- Well-written and organized paper

Weaknesses:

- No novelty in the method

- Missing details and insights for the experiments

- Lack of qualitative evaluation in sample efficiency compared to other RL.

----- Post rebuttal -----

In the rebuttal period, some reviewers’ concerns were resolved and turned into rather positive, and all the reviewers almost agreed that this paper should be accepted. Therefore, I conclude that the paper should be accepted. I strongly encourage the authors to take all the reviewers’ suggestions for the final version of the paper, especially clarifying the statement on the non-standard learning curves made with only one run for a deep RL algorithm to avoid any misleading to the readers.

**Best Paper Nomination:**

No

---

> ### Author Response · Authors · 2022-08-26
> **Rebuttal**
>
> We would like to thank the area chair for their summary of the paper, and for noting the impressive real-world performance and ease of use. We believe that our rebuttals have adequately addressed the weaknesses you noted, and highlight the potential impact of our work.
>
> **Novelty**: As a work in systems, our aim was to introduce the reader to the power of learned world models in the context of real world robot learning. As mentioned by R1: "Running Dreamer on real-world robots is novel to my best knowledge and is potentially of high impact." and R5: "To date, it is probably the most convincing example of a control policy trained in real life. The fact that all tasks can be solved with the same hyperparameters and minimal tuning is also remarkable." We agree with the reviewers, and we believe that world models will enable robotics researchers to solve complicated robotics tasks with high sample efficiency.
>
> **Sample efficiency compared to other RL techniques**: Thank you for pointing this out. It was noted by R1 and R3 that a better comparison of sample efficiency was required. As such, we created new plots to show the sample efficiency of Dreamer compared to other SOTA RL algorithms on 4 robots and tasks (https://imgur.com/a/TL3QgtT). Dreamer outperforms SAC, Rainbow, and PPO on the evaluated tasks, and matches the performance of DrQv2 on the pixel-based navigation task. We also ran an additional baseline to compare Dreamer to PPO + R3M encoder, as suggested by R4 (https://imgur.com/a/hWTKmTw). Even though PPO + R3M is aided with a pretrained visual representation, Dreamer still achieves higher sample efficiency when trained from scratch. We also ran a 24H training run, as suggested by R1, and demonstrated that after experiencing deteriorated performance due to changes in lighting conditions, Dreamer recovers and overcomes lighting condition changes faster the next time around (https://imgur.com/a/FXIo3CX). We will include these plots and descriptions in the final paper.
>
> **Missing details and insights**: Thank you for pointing out the lack of insights w.r.t. experiments. Please refer to our Official Comment titled "*Environment Details and Hardware Specification*", where we provide more details about each of the experiments. We hope that our detailed reviews have addressed the reviewer's concerns pertaining to experiment details.